# Enabling Value Co-Creation in Healthcare through Blockchain Technology

**DOI:** 10.3390/ijerph20010067

**Published:** 2022-12-21

**Authors:** Tiziana Russo-Spena, Cristina Mele, Ylenia Cavacece, Sara Ebraico, Carina Dantas, Pedro Roseiro, Willeke van Staalduinen

**Affiliations:** 1Department of Economics, Management, Institutions, University of Naples Federico II, Via Cintia Monte S. Angelo, 80126 Naplese, Italy; 2SHINE 2Europe, 3030-163 Coimbra, Portugal; 3TICE.PT—The Portuguese National ICT Cluster, Campus Universitário de Santiago (IT), 3810-193 Aveiro, Portugal; 4AFEdemy—Academy on Age-Friendly Environments in Europe, 2806 ED Gouda, The Netherlands

**Keywords:** value co-creation, blockchain technology, healthcare ecosystem

## Abstract

The COVID-19 pandemic highlighted the need to manage complex relations within the healthcare ecosystem. The role of new technologies in achieving this goal is a topic of current interest. Among them, blockchain technology is experiencing widespread application in the healthcare context. The present work investigates how this technology fosters value co-creation paths in the new digital healthcare ecosystems. To this end, a multiple case study has been conducted examining the development and application of blockchain by 32 healthcare tech companies. The results show blockchain technology adoption’s current and potential impacts on value co-creation regarding data and resource sharing, patient participation, and collaboration between professionals. Three main areas of activity emerge from the case studies where blockchain implementation brings significant benefits for value co-creation: improving service interaction, impacting actors’ engagement, and fostering ecosystem transparency.

## 1. Introduction

The COVID-19 pandemic can be considered the first real pandemic on a global scale in the digital age [1]. It created myriad challenges for health and care services worldwide and led to one of the biggest social crises of the century. As the pandemic progressed, digital health solutions emerged as the most promising tools to address many of these challenges. Although a digital transformation in healthcare had been envisioned for decades, and technological solutions had reached a certain level of maturity, their adoption remained low due to inertia and barriers such as lack of investment and human capital. The COVID-19 pandemic boosted the adoption process, testing the potential of digital technologies in healthcare within a few weeks and demonstrating their usefulness in addressing health crises [2]. The most utilized technologies were big data, robotics, the Internet of Things (IoT), artificial intelligence, and mobile applications [3]. These technologies were shown to be essential for improving the quality of healthcare services [4] when applied to telemedicine, home care, the tracking of contagions, and data management and sharing [5]. The pandemic also highlighted the importance of coordination and collaboration between different actors, including agencies, organizations, patients, institutions, and governments, even from other countries [6].

New and renewed value co-creation processes emerged throughout the pandemic, encapsulating the paradigm shift toward “patient-centered” healthcare [7]. In healthcare, value co-creation represents the “joint collaborative activities by parties involved in direct interactions, aiming to contribute to the value that emerges for one or both parties” [8] (p. 1523). Although it initially referred to the bilateral relationship between the service provider and patients, value co-creation’s meaning has been extended to include the multiple actors involved in the process [9]. Value co-creation occurs during healthcare encounters; it is influenced by information and data sharing, trust, role clarity, and actor experiences and is extended within the entire ecosystem of healthcare organizations [10]. In such a context, privacy, security, and interoperability are some of the most critical issues to address due to the sensitive nature of healthcare data. These issues include data accessibility, sharing, reliability, and trust. Currently, most healthcare ecosystems face significant obstacles or have deficiencies in these areas.

A growing demand, therefore, exists for decentralized, secure, and scalable databases, and blockchain offers a viable means of addressing these needs [11]. Consequently, this technology has attracted considerable interest in the healthcare community [12,13]. A blockchain is a decentralized, public digital ledger that records transactions across many computers. It consists of blocks of data linked with cryptographic protocols, thus making traceability possible. The data are stored in the distributed ledger and cannot be removed or altered without the knowledge and permission of the record’s creator and the network itself. The record cannot be changed retroactively without altering subsequent blocks [14]. Blockchain applications in healthcare are numerous. They can be used to trace pharmaceutical products, ensure the control and confidentiality of patient medical records, collect data and knowledge for research, and manage medical billing and bargaining [15,16,17]. Several researchers have analyzed Blockchain technology’s potential in managing pandemic situations [18,19,20]. Many potential uses have been identified: creating digital health passports [21]; making contact tracing more secure, immutable, and efficient [22,23]; facilitating clinical trial management [24]; easing data collection and distribution between stakeholders [25,26]; ensuring privacy in information sharing [27]; automating decision-making through smart contracts [28]; integrating to big data for data control [29]; allowing early detection of outbreaks [30,31]; fast-tracking drug delivery [30]; and enhancing e-government and supply chain management [26]. Given the various applications, investments are expected to overgrow in the next few years. According to a study conducted by Global Market Insights, the value of blockchain in healthcare will exceed $1.6 billion by 2027 [32]. However, much of the current research remains at the technical stage, with few studies providing clinical applications, thus highlighting the need to translate foundational blockchain technology into clinical use [33].

This paper analyzes how blockchain technology supports value co-creation to clarify blockchain’s current and future impact on the digital healthcare ecosystem. We employ a multiple case study investigation of 32 healthcare providers and technology companies. The analysis allows us to identify three main value co-creation activities within the healthcare ecosystem enabled by blockchain applications: improving service interaction, impacting actors’ engagement, and fostering ecosystem transparency. These activities are not standalone; they complement each other to increase value co-creation.

## 2. Literature Review

### 2.1. Healthcare Ecosystems and Value Co-Creation

The health crisis created by the COVID-19 pandemic highlighted the importance of coordination and collaboration between different stakeholders and systems in creating health value [6]. In such a context, enhanced healthcare service opportunities rely on a multidimensional construct of value co-creation that needs to consider different actors’ specific conditions of access and resource integration [34]. The ecosystem perspective introduced by the service-dominant logic theoretical framework [35] is applicable in this situation. Lusch and Vargo defined a service ecosystem as a relatively self-contained, self-adjusting system of resource-integrating entities connected by shared institutional logics and mutual value creation through service exchange. Within these ecosystems, value is co-created through the sharing and integration of resources between different actors connected by value propositions in an intense network of relationships [7]. This perspective allows for a systemic approach to co-creation that goes beyond the patient–provider dyad and involves other actors sharing, integrating, and creating the necessary resources to improve the quality of healthcare services [36,37].

Based on the ecosystem perspective, the literature recognizes different actors involved in value co-creation at varying levels of the healthcare ecosystem [38,39]. Most scholars focus on the micro level, where value is co-created through interactions between patients and other actors who share and combine resources to achieve the desired result [40,41]. The patient perspective is stressed by emphasizing value co-creation involving various activities around patients or their collaboration with other service network members, including family members, friends, other patients, healthcare professionals, and external communities [9,42]. Other researchers [9] have explored the practices of value co-creation at the meso, macro, and mega levels, which are realized through the respective sharing of resources between health organizations such as hospitals and clinics, professional associations, and public and insurance authorities, as well as government agencies, health funding bodies, regulatory bodies, and media [15]. By analyzing the interactions between several actors at different levels, it is also possible to understand their mutual influences and the dynamic evolution of the ecosystem.

The design of a healthcare ecosystem is represented as a complex integration of human-centered activities that are increasingly dependent on the real-time integration of various data and processes. In such an ecosystem, value co-creation can be fostered by the adoption of digital platforms [43] that offer a flexible and integrated structure to facilitate interactions, develop a shared vision, foster collaboration, and ensure the transparency of rules and the traceability of each actor’s contributions [44,45]. In addition, studies on smart health solutions demonstrate that interactive digital platforms favor patient engagement and active participation [43]. The connected technology can sense the conditions and surroundings, engaging patients in real-time data collection, continuous communication, and interactive feedback. Digital technologies also affect actors’ roles by widening resource accessibility and facilitating decision-making by integrating massive real-time data and multiple interactive responses [4]. They extend engagement in ways that help users adopt long-lasting changes and augment human actors’ agency to better monitor, update, and refine their decisions or execution efforts [4,46].

There is a standard agreement that value co-creation implies the need to consider the involvement of patients, caregivers, and other actors who require accessible, complete, and timely personal information [47]. There is also a demand for more significant interactions and coordination between service providers, specialist centers, physicians, other healthcare professionals, and patients [9]. This debate is still in progress. For example, the most effective way to fulfill the expectation for improved value co-creation in healthcare through multiple actors’ involvement is still under contention, especially considering that digital service healthcare is a highly complex network [47]. The literature advises that this is not only a matter affecting patients: it touches upon a broad number of actors and resources that could be better integrated and mixed. It progressively recognizes the nature of ecosystems in healthcare services [7].

### 2.2. Blockchain and Healthcare Ecosystems

Blockchain technology is a decentralized ledger that records data and transactions across a peer-to-peer structure [48]. Sets of a transaction form a block identified by a unique cryptographic hash, setting its time stamp. This hash code represents the first piece of data linked to the next block, forming a chain. The first application of blockchain technology was related to payments, emerging from the Bitcoin whitepaper [49] to provide a secure and anonymous way to transfer money between two parties without the existence of a centralized authority. Blockchain allows all parties involved in the network to access data and validate every transaction. All transactions are thus stored in an immutable manner [50]. This disruptive technology has been applied to supply chains [51], the energy sector, agri-food, and other business areas. It builds trust mechanisms for solving transparency and security issues related to information exchange, such as tampering [52].

Many studies on blockchain discuss its future potential in the healthcare industry. Most of them focus on the impact of blockchain on the management of electronic health records and personal health records to prevent unauthorized access or ensure data accountability and reliability [53,54,55]. According to some scholars, blockchain could improve many aspects of healthcare data management: access control, interoperability, secure provenance, and data integrity [56,57]. Within the healthcare ecosystem, the use of blockchain facilitates data sharing between healthcare providers, patients, and other stakeholders by supporting team-based care, continuity of care across institutional boundaries, identity management, and access control between different healthcare systems [58]. By making the collection, sharing, and storage of data captured through different devices (i.e., sensors, smart watches, etc.) more reliable, this technology can foster the development of mobile health and telemedicine [59,60,61]. Blockchain also increases patients’ control over their data by giving them greater responsibility and encouraging active participation [57]. It has been demonstrated that the digitalization of healthcare makes blockchain-based stakeholders’ interactions more secure and effective. For example, multiple actors participating in various digital interactions, such as patients with healthcare providers and physicians with patients, can enhance their access to vast amounts of secure information, promoting remote health monitoring, detection, and prevention of diseases or patients’ adherence to treatment.

Outside of the clinical context, blockchain has potential applications in other areas, such as biomedical research [62] and clinical trials, where this technology can enable the recruitment of patients and the secure and anonymous collection and sharing of their data.

Finally, within the field of drug logistics, blockchain can monitor and certify drug distribution in the pharmaceutical supply chain. By constantly monitoring the supply chain, the blockchain can prevent drug unavailability or solve logistics problems promptly by ensuring the supply goes where it is needed in the platform ecosystem [63]. The synergistic interaction of eHealth digital platforms and multilayer networks creates a proactive ecosystem that leverages healthcare blockchains, fostering massive adoption in a sensitive data industry where privacy concerns increasingly matter [64,65].

Many challenges in blockchain adoption have been discussed, with technological complications considered the critical factor affecting the future of blockchain-based solutions [54,66]. However, although significant technical challenges remain (e.g., privacy, scalability, interoperability), the technical issue might have been overemphasized. A primary reason may be that the early research on blockchain technology focused on developing novel algorithms, frameworks, and proofs of concept rather than analyzing its use in the operative context more seriously. The value-process-related aspects acquire importance mainly concerning a detailed acknowledgment of other potential benefits and challenges to its adoption. Studies prove that healthcare organizations can benefit from further investigation analyzing blockchain technologies for value co-creation.

## 3. Materials and Methods

This paper adopts a multiple case study method to investigate the application of blockchain for value co-creation in the healthcare ecosystem [67,68]. This approach involves an in-depth exploration of bounded phenomena, utilizing multiple data collection to systematically gather information on participants’ perspectives on the phenomenon within its natural context.

To select cases, we followed a snowball sampling technique. More specifically, the process started with a few cases identified through collaboration with members of the Italian Blockchain Association. These companies have adopted and developed healthcare blockchain solutions to improve their service provision and activities, transforming the relationship between the actors involved. Following the recommendation of Seawright and Gerring [68], we added cases that incorporate different combinations of IoT and other technologies in deploying blockchain solutions to achieve diversity.

To collect data, we used semi-structured interviews to encourage informants to provide insights into implementing and using blockchain technology. Interviews were stopped when we reached a saturation point [69]. Interview questions were aimed at identifying the following issues: the impacts of blockchain on interactions within the healthcare ecosystem; the actors involved; the eventual engagement of new actors in the value creation process after the adoption of the technology; changes in relationship management, and eventual extension of the ecosystem network. The information collected was complemented with data on the company’s features, leading solutions provided, clients, and applications from secondary sources, such as official company websites, videos, and reports [69]. We first analyzed internal case data and created a detailed overview of each case. To synthesize the information, we applied the data coding procedure recommended by Yin [67] and grouped the data into tables according to the following variables: (1) state of the progress of the developed solutions, (2) integration of blockchain with AI technologies, and (3) co-creation practices. We first used open coding to identify initial themes to analyze these data. Then, in a second step, we carried out an analytical data process, linking the verbatim analysis with text mining and lexical analysis. In the data coding process, we relied on thematic analysis. Instead of a mechanistic data-reduction approach, we sought to transform the raw data to a more conceptual level [70]. Our research approach used cross-case analysis to identify similar themes across cases. They included the actors involved, the value co-creation drivers, and features enabled by the technologies. To ensure research credibility (i.e., internal validity), the team submitted their interpretation to the scrutiny of the individuals on whom it was based and obtained their perspectives on its authenticity [67].

### Sample

A theoretical sample of 32 cases is analyzed (see Table 1). The final sample is composed of 11 Italian firms and 21 International companies. All Italian companies are private developers of technology solutions for several sectors, including healthcare. In some instances, they have created specific partnerships for projects exclusively dedicated to the healthcare sector, as in the case of In2Dafne, a project developed by a collaboration between the Dafne Consortium and Intesa Bank for supporting the healthcare supply chain and the distribution of pharmaceutical products across different regions. The leading blockchain applications offered by the Italian companies are related to the certification and traceability of vaccination and sanitization of workplaces, the traceability and exchange of medical records for telemedicine, and transparency of the supply chain for combating drug counterfeiting. The main customers are hospitals, private clinics, laboratories, drug distribution and logistics companies, and research institutions. Some blockchain solutions facilitate compliance with legal procedures for private companies. 

In the international scenario, we examined 13 US companies, two UK firms, and three European projects, with other cases from Russia, Estonia, and Japan. Most technology providers (17 in total) specialize in healthcare solutions, and most are technology experts in the exchange and certification of genomic data to advance research in rare diseases or the traceability of medical records for clinical trials and telemedicine. Other solutions are related to the use of blockchain within global health in terms of cryptocurrencies and health financing and insurance, supply chain management, identification, verification processes, telehealth, and misinformation. Many technology providers form consortia between health companies, ICT, and big corporations to organize the upstream ecosystem and speed up integrating different platforms developed to improve healthcare. In addition to providing solutions to businesses and end users, these providers cooperate with associations, chambers of commerce, and governments to extend the use of blockchain to the public sector.

## 4. Results

The cases analyzed show various applications of blockchain technology capable of supporting value co-creation in healthcare. Most blockchain solutions developed by these companies create platforms for tracking clinical data (Electronic health records, real-time parameters from medical devices, clinical laboratory analysis) to facilitate the development of personalized treatments and increase collaboration between physicians and patients. These applications are designed to monitor patients’ health conditions (including remotely) and offer real-time solutions. By facilitating the secure sharing of data between patients, physicians, and caregivers, the platforms enable service interactions that positively impact service quality and patient trust. Often these solutions are combined with AI tools to achieve an automated diagnosis, improve telemedicine services, process medical images, and data from wearable devices, prevent future diseases, and create a broader clinical picture of the patient. 

Three main value co-creation activities can be detected by analyzing the cases: improving service interaction, impacting actors’ engagement, and fostering ecosystem transparency. The actors involved, the blockchain features, and value co-creation drivers have been identified for each. Table 2 provides a synthesis.

### 4.1. Improving Service Interactions

Our findings highlight the ability of blockchain technology to foster interactions between different actors in the healthcare ecosystem. In many of the analyzed cases, it is possible to detect how the adoption of blockchain for patient data management and insurance payments, supply chain management, and telemedicine extends the joint sphere of value co-creation for the actors involved. Technology providers, physicians, healthcare professionals, researchers, pharmacists, and patients all interact by sharing data and resources securely and effectively due to transaction traceability, privacy protection, increased interoperability, integrity, accessibility, and coordination of shared resources. Thanks to blockchain technology, patient access to data can be better managed and controlled. For example, MyHealthMyData, a Horizon 2020 EU project, developed a mobile app to manage the exchange of sensitive data from different sources (medical records, mobile apps, IoT). Users have ultimate control over the data as they can set their consent options and decide how long data are available for usage purposes and what type of actors (patients, private or public institutions) can access it. The same is for patients’ data, which can be controlled via the access options through the app. 

In many cases, the result is the achievement of patient-centered care that provides real-time solutions based on improved service quality and real time service. For example, through the blockchain-based Robomed network, healthcare providers and patients can interact using smart contracts. Within the network, a mobile application allows patients to receive telemedicine consultations and exchange EHRs; healthcare providers can record and use diagnostic processes and monitor performance metrics, and companies can monitor and verify the health status of patients and adhere to clinical guidelines for the provision of healthcare services. The aim is to restore a patient’s state of health with the least expenditure of time and money and to provide efficient medical performance based on feedback that other participants in the network can consult. Similarly, the Blockcom messaging platform, provided by Reply SPA, enables certified messages to be sent between biomedical devices and service providers, ensuring the real-time exchange of accurate and verifiable data on patient parameters between network participants.

Many analyzed cases show how blockchain adoption can support patient and doctor interactions. In the UK, the decentralized Medicalchain platform was introduced to collect patient records in a single database that guarantees privacy through encryption and asymmetric keys. Within the platform, patients grant professionals access to their data for clinical trials through smart contracts and receive a reward. Patients also access the platform to monitor their health status or to seek medical advice and opinions, while doctors and pharmacies can record transactions and notes. Every interaction is recorded on the blockchain as a transaction, thus becoming traceable and transparent. In the same vein, the TrustedChain blockchain platform, developed by the Italian company Ifin Sistemi, makes it possible to collect and share patients’ clinical data for scientific, statistical, and commercial purposes, guaranteeing security and privacy. Through the platform, patients, research organizations, and hospitals can interact securely and in an atmosphere of mutual trust. Sensitive data are not visible to network nodes, and patient identity and consent management are automated and secure.

Other solutions have been designed to manage interactions between actors such as professionals, manufacturers, and distributors. For instance, ProCredEx created a secure and reliable platform based on the private blockchain Corda, where physicians can exchange and verify their data and accreditations as healthcare professionals. The company has created an extensive and reliable network of accredited healthcare professionals, benefiting the entire healthcare ecosystem and leading healthcare organizations to use the platform to manage and improve cost-intensive and time-consuming clinical accreditation processes. Another example is the US company Chronicled. They developed the MediLedger network, which combines a secure peer-to-peer messaging system and a decentralized blockchain system for communication between manufacturers, wholesalers, and purchasing organizations in the healthcare sector. The platform ensures alignment, enabling a unified view of transaction data between trading partners (e.g., contracts, customers, prices, products), reliability and accuracy through secure data transactions, and speed and automation of transactions that take place without the need for a third-party guarantor. This way, trading partners are aligned on the latest contract status, and disputes and charging requests can be resolved quickly, eliminating the risk of delays and inaccuracies for wholesalers and improving customer price accuracy.

### 4.2. Impacting Actors’ Engagement

The analysis of the selected case studies reveals the role of blockchain in fostering the engagement of patients and other actors in healthcare processes. In most cases, it is apparent that this technology enables the participation of patients, technology providers, researchers, and other actors in the value co-creation process through the sharing of sensitive information in a protected and secure environment in which the use of their data is always traced. 

Coral Health, for example, offers patients a blockchain-based application where they can enter their data and be assisted by technology in managing drugs and prescriptions. Patients can also share their data with doctors and caregivers for further support and assistance. The role of blockchain is to facilitate this process, ensuring that shared data are fully protected through encryption and giving the patient complete control over personal information.

Iryo Network, a clinical data storage platform, offers a similar solution designed for healthcare providers rather than patients. The network provides software with an anonymous query interface through which doctors can digitize patient records, set appointment reminders, send follow-up notes to patients, schedule check-ups, and check payment status. This way, the tool helps doctors engage patients in the care pathway and stimulate active and responsible participation. The platform uses blockchain authorization controls for patients’ registration and access and AI to generate tokens that accelerate the receipt of consent from end users.

Blockchain platforms help research organizations engage patients for resource integration and collaboration while ensuring secure digital transactions and anonymity when applied to clinical and scientific research and clinical trials. One example is the Gene-Chain platform developed by EncryGen, which offers patients incentives for their active participation through cryptocurrencies. The platform was designed to manage digital transactions involving genomic information in DNA research. Patients are incentivized to upload their DNA profile on the platform by setting the price at which they are willing to sell it and deciding with whom to share their data, which is protected by anonymity. Data buyers can search the Gene-Chain marketplace for the profile that best fits their scientific project and purchase it via “$DNA tokens” exchangeable for Bitcoin and other cryptocurrencies. All transactions are recorded immutably. 

Another critical solution for engaging individuals to participate in the advancement of scientific research is Nebula Genomics. This platform allows researchers to obtain a person’s entire genome sequence at a low cost. After purchasing a kit and submitting a test, users access the Nebula Genomics website and receive the results. Users can protect their data through blockchain technology during kit registration, and consent is securely and transparently managed on an immutable public ledger. If the user accepts the usage policy, Nebula Genomics will generate new genetic reports based on the latest scientific findings. Before the data are analyzed to create a report, the user’s authorization and compliance with data privacy regulations are verified. The encrypted genome files are then uploaded into a secure enclave, where Nebula Genomics executes the code to compile the new report. In this way, the system incentivizes the patient to share resources with other actors in the ecosystem, taking an active role in the value co-creation process.

### 4.3. Fostering Ecosystem Transparency

A third area where blockchain encourages value co-creation is related to ecosystem transparency. Blockchain technology enables secure data verification by storing immutable information that prevents tampering. It, therefore, creates a trusted environment because every data and action is verifiable and cannot be manipulated. Transparency also allows an entire ecosystem, such as a supply chain, to be visible, which makes it easy to identify inefficiencies and clarify responsibilities, thereby activating a virtuous trust circle between stakeholders. Because of its decentralized nature and the cooperative sharing of information among the participants, blockchain facilitates the constitution of a consolidated group with a specific shared aim. For instance, Avaneer Health, a consortium of firms managed by IBM that contributes to healthcare improvement, ties funders, suppliers, and other organizations together to build a sustainable ecosystem supporting the use of a wide range of platforms. Avaneer’s technology is designed to improve the patient experience, especially in resolving administrative issues and sharing personal data. Transparency and trust between independent actors also allow for control of the quality certifications of doctors and physicians so that patients can receive adequate and specialized medical care. 

Our findings also reveal that transparency positively impacts coordination mechanisms for more effective data management processes. Blockchain provides a complete overview of patients’ clinical status by integrating and verifying all clinical data from multiple sources into a single database. It influences the perception of the motivations governing the other party’s action (intentions) and the perception of his/her competence to act in favor of the patient’s interest. For example, Pharmaledger developed a platform to serve the entire pharmaceutical ecosystem using a scalable and sustainable architecture to achieve efficient decentralization. The aim is to create a system in which all clinical trials and screenings from the entire pharmaceutical industry converge to provide the best treatment solutions for patients. The platform creates a shared register accessible to network participants—subject to prior authorization—which allows them to consult the history of treatments validated by medical firms. Further, blockchain technology assures data access and increases efficiency in healthcare coordination while preventing tampering and the unauthorized use of data. For example, Patientory Inc. developed a platform for gaining access to information among all stakeholders in the healthcare ecosystem, removing processes currently hindering coordinating healthcare. With the implementation of Patientory’s blockchain infrastructure, healthcare professionals can minimize access control breaches in the system, improving service coordination and facilitating the handling of complaints in real-time.

Ecosystem transparency reflects a superior value co-created by many actors, who are encouraged to collaborate, communicate, and share resources in a visible environment provided by blockchain features. In this way, the different actors are incentivized to participate in the co-creation of a mutual benefit in a win-win logic.

Blockchain also prevents unfair practices thanks to the more effective and transparent actors’ cooperation and participation, as demonstrated by the fight against drug counterfeiting. The solution created by FarmaTrust uses blockchain to create an immutable register for recording supply chain operations to ensure the safety and authenticity of pharmaceutical products, developing an end-to-end transparency solution. The main objectives are improving patient health, eliminating counterfeit products, increasing efficiency, and developing advanced tools for data analysis. Pharmacies and patients can verify a product’s authenticity by scanning the information placed on the label using a QR code. If a regulator or manufacturer labels the product unfit or discovers inconsistencies in its transaction history, the consumer is instantly contacted to return the product. The FarmaTrust solution can increase the coverage and effectiveness of automated checks and payment tracking. Similarly, the platform In2Dafne, developed by the Italian Consorzio DAFNE, ensures visibility to all actors in the supply chain using blockchain, making data related to drug stocks visible. 

## 5. Conclusions and Implications

In this paper, we developed a framework that considers how blockchain affects value co-creation in healthcare ecosystems. 

Previous studies have discussed the role of technology as a facilitator of value co-creation processes in healthcare, providing evidence about the specific adoption of blockchain technology. Our framework adds to this research by offering a comprehensive view of the stakeholders participating in value co-creation activities in the healthcare ecosystem [7]. Our framework detects three main value co-creation activities that blockchain supports: improving service interaction, impacting actors’ engagement, and fostering ecosystem transparency. These activities are not exclusive; there is a dynamic interplay between them. The blockchain supports provider-patient and other stakeholder interactions easily within the service process by making information more accessible, exploitable, and appropriate to specific uses. Data are fundamental for collaboration and value co-creation. By allowing traceability, blockchain clarifies the roles and contributions of each actor, increasing their incentive to co-create and providing an environment that stimulates a more engaging experience. At each interaction, such engagement reinforces value creation, making it visible and traceable. By activating more effective means to coordinate expectations and interaction in relationships, blockchain allows a new form of trust to emerge as an organizing principle in the ecosystems (see Figure 1).

From a practical point of view, our results show that despite the various benefits of using blockchain in healthcare, its implementation still needs to be improved. Several actions are required to promote the use of these technologies to support value co-creation processes. First, effective educational and training programs should be planned to increase patients’ health literacy and empowerment, making them aware of their role as resources in the healthcare system and encouraging participation in the co-creation of health value using new technologies. At the same time, companies and health professionals should be supported in using these technologies through funding and staff training. Finally, the role of government is crucial in promoting shared solutions at the macro level, providing appropriate regulations and guidelines for using these technologies, and adapting current data protection laws to the emerging possibilities of data use offered by these solutions. Blockchain’s application in healthcare currently lacks regulatory and formal guidance, which could limit its use. The transparency, accessibility, interoperability, and decentralization of data enabled by blockchain are, at the same time, the most in-demand features in healthcare—and the most problematic. As health data is highly sensitive, it is also heavily regulated to ensure patient privacy. If designed and used in accordance with regulatory frameworks such as GDPR and HIPAA, blockchain could help mitigate the risk of privacy breaches in healthcare data. However, according to the study by Haselgren et al. [15], no available blockchain platform can demonstrate compliance out of the box. There is only the international standard, ISO 22739:2020 Blockchain and Distributed Ledger Technologies—Vocabulary, which provides foundational terms featuring definitions for blockchain. Other standards, such as IEEE 2140.1-2020 and IEEE 2140.5-2020, have been provided for the specific use of cryptocurrency. ISO has also developed a series of technical reports that differ from standards as they only give an overview of the issues and practical concerns related to the smart contract, security management of data and privacy, and personally identifiable information protection.

## 6. Limitation and Further Research

This study has some limitations, mainly related to the method. The qualitative approach is exposed to a potential bias associated with the self-selection of research. The multiple case study approach does not allow a generalization of results; therefore, future studies should expand the dataset. In addition, this study solely adopts the healthcare providers’ and user companies’ approaches.

Further research should widen ecosystem perspectives or adopt the patient-centered approach to provide innovative ways to foster actor engagement and co-creation opportunities. Future research could also investigate the individual blockchain-based solutions and their effects more deeply through a comparison based on the development of indicators measuring aspects such as interoperability, engagement, satisfaction, performance, etc.

Moreover, future studies could further investigate how to use blockchain to support patients during pre- and post-inpatient periods and better educate them in managing their data. Giving patients the ownership of their health and personal data is not without risks; every party, agency, channel, or node that accesses a set of data would have some claim to it, but also some rights and responsibilities that they have to abide by when accessing that data. 

In addition, based on the results of this study and considering the transversality of this technology, it would be interesting for future research to investigate how blockchain could improve the relationship between different ecosystems that revolve around the health sector, supporting the co-creation of value between them. One example would be to study the use of blockchain in medical tourism to integrate health, tourism, and government ecosystems. The medical tourism phenomenon is, in fact, rapidly expanding. Future research could investigate the co-creation of value in blockchain-based platforms that assist customers in finding better solutions in their medical tourism experience, offer a patient-centric information model that provides a comprehensive assessment of their health, and contribute to the growth of the destination country. Another future insight to highlight is the possibility of increasing the engagement of different actors by developing healthcare communities with tokens.

Finally, the digitalization of healthcare ecosystems could represent a needed prerequisite of blockchain. In other words, the lack of digitalization limits the application of blockchain. Further research could go in-depth into analyzing the degree of digitalization within healthcare ecosystems and how it affects blockchain adoption.

## Figures and Tables

**Figure 1 ijerph-20-00067-f001:**
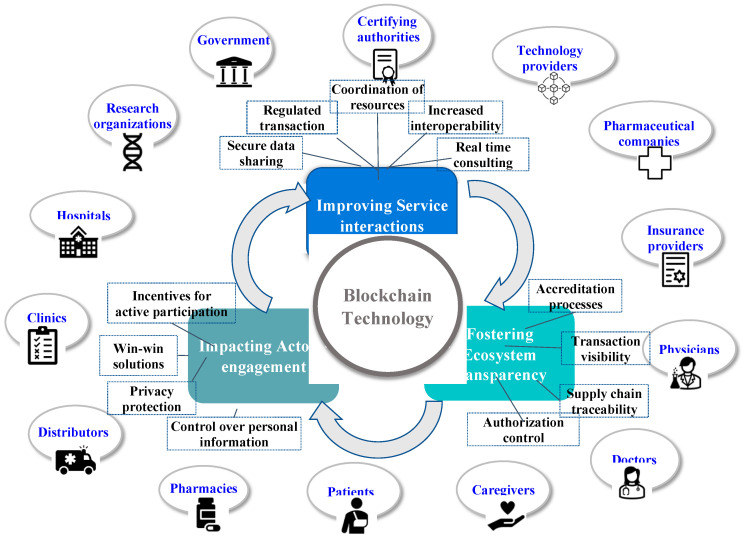
Blockchain-based value co-creation in Digital Health Ecosystem.

**Table 1 ijerph-20-00067-t001:** Companies investigated in this research.

Company	Blockchain Solution(s)	Applications
Akiri	Akiri Switch	Traceability and exchange of medical records
Avaneer Health		Traceability and transparency of medical records
B2Lab	SaniStory	Certification and traceability of vaccinations
Blockchain Italia	My Health Passport	Certification and traceability vaccination
BurstIQ	LifeGraph	Certification and traceability of health records for telemedicine
Chronicled	MediLedger Network	Certification and traceability of commercial agreements
Consulcesi Tech	Futura Stem Chain	Certification and traceability of stem cells
Coral Health		Certification and traceability of health records
CrystalChain	Blockpharma	Certification and traceability of drug supply chain
Embleema	HIVE	Certification and traceability of health records for clinical trials
EncrypGen		Certification and exchange of genomic data for advanced research
Engineering	InteropEHrate	Traceability and exchange of medical records
EZ Lab		Certification and traceability of COVID-19 diagnostic kits
Factom Inc.		Certification and traceability of health records
FarmaTrust	Zoi	Certification and traceability of drug supply chain
GuardTime	Vaccine Guard	Certification and traceability of vaccination and records for clinical trials
HealthVerity	HealthVerity Consent	Certification and traceability of patients’ consent
Ifin Sistemi	TrustedChain	Certification and traceability of health records
In2Dafne		Traceability of drug supply chain
InfoCert S.p.A.	Health Checker	Certification and traceability of digital identity
Iryo Network	Iryo Moshi Practice Management Software	Traceability and transparency of medical records for telemedicine
Medicalchain		Certification and traceability of health records for telemedicine
My Health My Data		Certification and exchange of health records
Nebula Genomics		Certification and exchange of genomic data for advanced research
Patientory Inc	Patientory	Certification of medical records and insurance payments
Pharmaledger		Traceability of medical records and drugs to contrast counterfeiting
ProCredEx	Corda	Traceability and transparency of digital professional identity
Reply SPA	Blockcom	Certification and exchange of health records for telemedicine
Robomed Network	Robomed EHR	Certification and traceability of health records for telemedicine
SimplyVital Health	Nexus Health Platform ConnectingCare	Traceability of health records and management of health care claims
Smartree	FidesChain and EventChain	Traceability and transparency of the supply chain
Var Group	Blockit	Traceability and exchange of medical records for telemedicine

**Table 2 ijerph-20-00067-t002:** Value co-creation enabled by blockchain.

Activities	Actors/Organizations	Blockchain Features	Drivers
Improving service interactions	Technology providers, physicians, caregivers, researchers, pharmacists, patients	Secure data sharing and management, privacy, transparency, traceability, participation	Patient-centered care, increased trust, improved service quality, real-time solutions, resource sharing, cooperation, active patient role
Impacting actors’ engagement	Technology providers, patients, researchers, physicians	Secure data sharing, transparency, security of the digital transaction, traceability, anonymity, participation incentives, immutability	Information sharing, active patient role, patient as an operant resource, effective collaboration, shared cultural vision
Fostering ecosystem transparency	Technology providers, pharmaceutical companies, distributors, hospitals, patients, governments, regulators, insurance providers	Secure data sharing, trust, transparency, traceability, inventory management, stock efficiency, prevention of drug counterfeiting, anonymity	Collaboration, dialogue, information accessibility, resource sharing, increased trust, improved service quality

Source: our elaboration.

## Data Availability

Not applicable.

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
