# Peer review of "Enabling Value Co-Creation in Healthcare through Blockchain Technology"

_ijerph, 2022, doi:10.3390/ijerph20010067_

Round 1

Reviewer 1 Report

GENERAL COMMENT

Dear Authors,

thank you for allowing me to read this interesting work. I believe that it is imperative to approach health-related topics considering the enormous technological changes that are taking place.

This paper starts by considering the significant challenges posed by the pandemic in handling complex relationships in the healthcare context. The pandemic has undoubtedly acted as a digital accelerator, enabling innovations such as blockchain to be applied in healthcare. Indeed, these tools have allowed for even more movement in the direction of 'patient-centred' changes and management. This paper has analysed 32 case studies, highlighting the widespread use of blockchain for different purposes, involving several actors simultaneously, increasing their involvement and the transparency of the services offered. The work sheds light on the benefits of using such blockchains and provides a glimpse into the future of their applications.

INTRODUCTION

The introduction is well-written, offering the reader references to the article's main topic. The only suggestion I would make is to add references and information about blockchain. Even if it is a spreading technology, it is reasonable to consider that not all future readers know blockchain and its functioning very well.

METHOD

The methodology is well described.

RESULTS AND CONCLUSION

The results are presented appropriately. Although I understand the choice of having reported only results and conclusions, I think it might be helpful to revise these parts, creating a 'discussions and conclusions' paragraph, then separating the part on limitations and future directions. This would make it possible to focus on the one hand on the conspicuous results that have emerged, which are made readable by the included tables and figures, and on the other hand, to read the results in the light of the comparison with the current state of the art. I repeat that this is only a suggestion and not an imperative. The authors are free to choose whether to adopt this subdivision or not.

IMPLICATION

The authors add some relevant future implications of the designed framework.

FINAL COMMENT

I suggest rereading the manuscript and using coherently in the text "BC" or "blockchain". There is some refuse in the text, for example, double spaces. I hope my comments and suggestions will help improve the already good quality of the manuscript.

Author Response

Dear Reviewer,

We want to thank you. Your comments and suggestions helped us to improve the content and the flow of the paper.

By following your  suggestions, as well as the suggestions of the reviewer n.2,  we summarized them in the next bullet point list

  1. We worked on the introduction to improve the problem statement and added some details on how blockchain work
  2. We worked on method and r to better describe the sample
  3. We introduced limitations and created two different paragraphs, one we titled “conclusion” and the second titled “limitation and further research”.

Please, find here below the detailed responses to all your comments.

1. Comment

The introduction is well-written, offering the reader references to the article's main topic. The only suggestion I would make is to add references and information about blockchain. Even if it is a spreading technology, it is reasonable to consider that not all future readers know blockchain and it's functioning very well.

Authors’ reply: Thank you for the comment. We added references and provided more details on blockchain. Please consider the new sentences in the introduction.

A blockchain is a decentralized, public digital ledger that records transactions across many computers. It consists of blocks of data linked with cryptographic protocols, thus making traceability possible. The data are stored in the distributed ledger and can no longer be removed or altered without the knowledge and permission of the record's creator and the network itself. The record cannot be changed retroactively without altering subsequent blocks [14].

2. Comment

The results are presented appropriately. Although I understand the choice of having reported only results and conclusions, I think it might be helpful to revise these parts, creating a 'discussions and conclusions' paragraph, then separating the part on limitations and future directions. This would make it possible to focus on the one hand on the conspicuous results that have emerged, which are made readable by the included tables and figures, and on the other hand, to read the results in the light of the comparison with the current state of the art. I repeat that this is only a suggestion and not an imperative. The authors are free to choose whether to adopt this subdivision or not.

Authors’ reply: Thank you for the comment. We introduced limitations and created two paragraphs, one titled “conclusion” and the second titled “limitation and further research.”

3. Comment

I suggest rereading the manuscript and using it coherently in the text "BC" or "blockchain." There is some refuse in the text, for example, double spaces. I hope my comments and suggestions will help improve the already good quality of the manuscript.

Authors’ reply: thank you. We did it

Reviewer 2 Report

Summary

This paper presents an interesting method of value co-creation within the healthcare sector, providing examples of blockchain solutions developed to perform specific activities. However, some details have not been introduced in depth, which make it difficult to read in some places. In order to improve the paper, the following suggestions and opinions are put forward.

General comments

The problem statement could be better introduced by giving a wider context to the research field. For instance, the covid-19 pandemic is only mentioned, but the issues related to such a situation could be better analysed to underline how the structured use of blockchain (and other) technology could have a positive impact on eventual future pandemic situations.

The literature review could be improved by looking at previous research tackling the topic on a wider perspective. The number of references should be increased, in order to provide more approaches and points of view on the discussed topic.

It could be interesting to introduce references to the International Standards for blockchain (e.g. ISO 22739:2020). How these standards can be integrated with legislation within the healthcare system? Is this a “no man's land” from a legislative perspective?

The investigated companies could be better introduced and analysed including an additional paragraph. This way the two-pages table could be synthesized (the long table does not help the reading flow), and the different Bc solutions could be better compared among each other. It is not clear the context in which these companies are working (country, public/private etc.).

The methodological process is not clear; how data from questionnaires and data from secondary sources was merged? Did they have the same weight in the analysis?

An overall comparison of selected Bc solutions would be interesting, by using a set of quantitative and qualitative indicators, to give more relevance to the results.

Other specific comments

In line 28 it is mentioned that covid-19 “wholly transformed the healthcare ecosystem digitally”. Is that true? Which reference inspired such a sentence? Probably the whole pandemic situation highlighted current issues related to the lack of digitalization within the healthcare ecosystem.

Lines 30-32: The connection between these two sentences is not clear; are technologies introduced to give a context for blockchain? If yes, a brief analysis could be useful to explain why the authors decided to focus on blockchain.

Line 35: formatting issue

Lines 49-50: this sentence is difficult to read

Line 54: Maybe here more references could be added, since it is said that “blockchain applications in healthcare are various etc.”

Line 55-58 What is the reference for the value of blockchain predicted in 2025?

In line 61-62 it is mentioned that “this paper analyses how blockchain technology supports value co-creation” but, at the current state, is it possible to state that healthcare systems lack digitalization, which represents the prerequisite of blockchain? Is this limiting the application of blockchain?

Table 2: It would be useful to better introduce stakeholders (actors), defining interest and power of different actors both in the value co-creation and blockchain process. What is the difference between ‘research organizations’ and ‘researchers’? If no difference, why are they called in a different way? Hospitals are considered one of the actors, but in what sense? From an infrastructural point of view? Or considering people working there?

Line 194: In the analysed tools, how is data sharing managed? For instance, patients cannot access all kind of data undoubtedly, so how is accessibility managed?

Author Response

Dear Reviewer,

We want to thank you. Your comments and suggestions helped us to improve the content and the flow of the paper.

By following your suggestions, as well as the suggestions of reviewer n.1,  we summarized them in the next bullet point list.

  1. We worked on the introduction to improve the problem statement and added some details on how blockchain work
  2. We worked on method and r to better describe the sample
  3. We introduced limitations and created two different paragraphs, one we titled “conclusion” and the second titled “limitation and further research”.

Please, find here below the detailed responses to all your comments.

1) Review comments: The problem statement could be better introduced by giving a broader context to the research field. For instance, the covid-19 pandemic is only mentioned. Still, the issues related to such a situation could be better analyzed to underline how the structured use of blockchain (and other) technology could positively impact future pandemic situations.

Authors’ reply: We did it. We revised the introduction and added the following sentences.

“The COVID-19 pandemic can be considered the first real pandemic on a global scale in the digital age [1]. It created many challenges for health and care services worldwide and led to one of the biggest social crises of the last century. Digital health solutions were instantly identified as the most promising tools to address these challenges. However, although digital transformation in healthcare had been planned for decades and technological solutions reached a certain level of maturity, their adoption remained low due to inertia and some barriers, such as investment and human capital. The COVID-19 pandemic boosted the process, testing the potential of digital technologies in healthcare within a few weeks and demonstrating their usefulness in addressing health crises [2]. The most used technologies were big data, robotics, the Internet of Things, artificial intelligence, and mobile applications [3]. These technologies have been demonstrated to be essential for improving the quality of healthcare services [4] when applied to telemedicine, home care, tracking of contagions, and data management and sharing [5]. To achieve effective results, the pandemic also highlighted the importance of coordination and collaboration between different actors, including agencies, organizations, patients, institutions, and governments, even from different countries [6].”

2) Review comments: The literature review could be improved by looking at previous research tackling the topic from a broader perspective. The number of references should be increased to provide more approaches and points of view on the discussed topic.

Authors’ reply: thank you. We revised and expanded the literature review paragraphs. Please see the new sub-paragraphs 2.1 e 2.2

3) Review comments: It could be interesting to introduce references to the International Standards for blockchain (e.g., ISO 22739:2020). How can these standards be integrated with the legislation within the healthcare system? Is this a “no man's land” from a legislative perspective?

 Authors’ reply: Thank you for the comment. We did it and added this point in paragraph 5. Conclusion and implication. Please see the following sentences. “Blockchain application in healthcare lacks regulatory and formal guidance, which could limit its use. The transparency, accessibility, interoperability, and decentralization of data enabled by blockchain are, at the same time, the most demanded features in healthcare and those creating the most problems. As health data is highly sensitive, it is also heavily regulated to ensure patient privacy. If designed and used in accordance with regulatory frameworks such as GDPR and HIPAA, blockchain could help mitigate the risk of privacy breaches in healthcare data. However, according to the study by Hasselgren et al. (2020), no available blockchain platform can demonstrate compliance out of the box. There is only an international standard, ISO 22739:2020 – Blockchain and Distributed Ledger Technologies – Vocabulary, which provides foundational terms featuring definitions for blockchain. Other standards, like IEEE 2140.1-2020 and IEEE 2140.5-2020, have been provided for the specific use of Cryptocurrency. ISO has also developed a series of technical reports that differ from standards as they only give an overview of the issues and practical concerns related to the smart contract, security management of data and privacy, and personally identifiable information protection.

4) Review comments: The investigated companies could be better introduced and analyzed including an additional paragraph. This way, the two-pages table could be synthesized (the long table does not help the reading flow), and the different Bc solutions could be better compared among each other. It is not clear the context in which these companies are working (country, public/private etc.).

Authors’ reply: We did it. We added a new sub-paragraph, 3.1. we titled Sample to provide more details on the investigated companies.

5) Review comments: The methodological process is not clear; how data from questionnaires and data from secondary sources were merged? Did they have the same weight in the analysis?

Authors’ reply: We clarified the methodological process. Please consider the new sentences in the method section as reported in the following: “To collect data, we used semi-structured interviews to encourage informants to provide insights into implementing and using blockchain technology. Interviews were conducted and stopped when we reached a saturation point [84]. Interview questions were aimed at identifying the following issues: the impacts of blockchain on interactions within the healthcare ecosystem; the actors involved; the eventual engagement of new actors in the value creation process after the adoption of the technology; changes in relationship management, and eventual extension of the ecosystem network. The information collected was complemented with data on the company’s features, leading solutions provided, clients, and applications from secondary sources, such as official company websites, videos, and reports [84]. We first analyzed internal case data and created a detailed overview of each case. To synthesize the information, we applied the data coding procedure recommended by Yin [82].and grouped the data into tables according to the following variables: 1) state of the progress of the developed solutions, 2) integration of blockchain with AI technologies, and 3) co-creation practices. We first used open coding to identify initial themes to analyze these data. Then in a second step, we carried out an analytical data process, linking the verbatim analysis with text mining and lexical analysis. In the data coding process, we relied on thematic analysis. Instead of a mechanistic data-reduction approach, we sought to transform the raw data to a more conceptual level [86].  Our research approach uses cross-case analysis to identify similar themes across cases. They include the actors involved, the value co-creation drivers, and features enabled by the technologies. To ensure research credibility [i.e., internal validity], the team submitted the interpretation to the scrutiny of the individuals on whom it was based and obtained their perspectives on its authenticity [82].

6) Review comments: An overall comparison of selected Bc solutions would be interesting by using a set of quantitative and qualitative indicators to give more relevance to the results.

Authors’ reply: We agree with your comments, but it is not the focus of this paper. However, we consider the importance of identifying quantitative and qualitative indicators, and we added these points as a limitation of this study that further research can try to address. Please consider the following sentences in the “Limitation and further research section.” 

“The study has some limitations, mainly related to the method. The qualitative approach is exposed to a potential bias associated with the self-selection of researchers. The multiple case study doesn’t allow a generalization of results. Future studies should expand the dataset. We focused on healthcare providers' and user companies' approaches. Further research should widen ecosystem perspectives or adopt the patient-centered approach to provide innovative ways to foster actor engagement and co-creation opportunities. In this paper, we focused on the value co-creation processes enabled by blockchain technology. Future research could investigate the single blockchain-based solutions and their effects more deeply through a comparison based on the development of indicators measuring aspects such as interoperability, engagement, satisfaction, performance, etc.”

7) Other specific comments

7.1 Review comments: In line 28, it is mentioned that covid-19 “wholly transformed the healthcare ecosystem digitally.” Is that true? Which reference inspired such a sentence? The pandemic probably highlighted current issues related to the lack of digitalization within the healthcare ecosystem.

Authors’ reply: We agree with your comment. We reshaped the sentence.

7.2 Review comments: Lines 30-32: The connection between these two sentences is unclear; are technologies introduced to give a context for blockchain? If yes, a brief analysis could be useful to explain why the authors decided to focus on blockchain.

 Authors’ reply: Thank you for your suggestion. We reshaped the sentence.

7.3 Review comments: Line 35: formatting issue

Authors’ reply: we revised the text.

7.4 Review comments: Lines 49-50: this sentence is difficult to read

Authors’ reply: Thank you, we revised the text.

7.5 Review comments: Line 54: Maybe here more references could be added since it is said that “blockchain applications in healthcare are various etc.”

Authors’ reply: we did it

7.6 Review comments: Line 55-58 What is the reference for the value of blockchain predicted in 2025?

Authors’ reply: we added the reference to the list

 7.7 Review comments: In lines 61-62 it is mentioned that “this paper analyses how blockchain technology supports value co-creation,” but, at the current state, is it possible to state that healthcare systems lack digitalization, which represents the prerequisite of blockchain? Is this limiting the application of blockchain?

Author's Reply: Dear review, we understand your point. However, digitalization is optional for blockchain, even if the two technologies appear more and more linked. We provide to explain this point better and added also this point as a limitation in the Future research paragraph. Please see the following sentences:

“Finally, the digitalization of healthcare ecosystems could represent a needed prerequisite of blockchain. Thus the lack of digitalization could be supposed to limit the application of blockchain. Further research could go in-depth in analyzing the digitalization degree of healthcare ecosystems and how it affects blockchain adoption”.

7.8 Review Comment : Table 2: It would be useful to better introduce stakeholders (actors), defining the interest and power of different actors in the value co-creation and blockchain process. What is the difference between ‘research organizations’ and ‘researchers’? If there is no difference, why are they called in a different way? Hospitals are considered one of the actors, but in what sense? From an infrastructural point of view? Or considering people working there?

Authors’ reply: thank you for the comment. We revised the table and clarified our meanings. We included in the second column both the actors as individuals and the organization in which they work. We mainly consider the organization when we deal with the ecosystem level and discuss the third activity based on the ecosystem’s trust.

7.9 Review Comment : Line 194: In the analysed tools, how is data sharing managed? For instance, patients cannot access all kinds of data undoubtedly, so how is accessibility managed?

Authors’ reply: Thank you for the suggestion. We try to explain better how blockchain supports the patient's access to data. For this reason, we introduced the new sample of MyHealthMyData in subparagraph 41. Please see the following sentences.

“Thanks to blockchain technology, the patient's access to data can be better managed and controlled. For example, MyHealthMyData, a Horizon 2020 EU project, developed a mobile app to manage the exchange of sensitive data from different sources (medical records, mobile apps, IoT). Users have ultimate control over the data as they can set their consent options and decide how long data are available for usage purposes and what type of actors (patients, private or public institutions) can access it. The same is for patients' data, who can control the access options through the App.

Round 2

Reviewer 2 Report

The revisions added value to the paper, thank you for the answers.